

# Laser spectroscopy of light muonic atoms and the nuclear charge radii

**Aldo Antognini[1,2*], Franz Kottmann[1,2] and Randolf Pohl[3]**

**1** Institute for Particle Physics and Astrophysics, ETH Zurich, 8093 Zurich, Switzerland
**2** Paul Scherrer Institute, 5232 Villigen–PSI, Switzerland
**3** QUANTUM, Institut für Physik & Exzellenzcluster PRISMA[+], Johannes
Gutenberg-Universität Mainz, 55099 Mainz, Germany

⋆ aldo@phys.ethz.ch

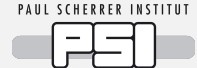

## Abstract

The energy levels of hydrogen-like atomic systems are shifted slightly by the complex structure of the nucleus, in particular by the finite size of the nucleus. These energy shifts are vastly magnified in muonic atoms and ions, *i.e.* the hydrogen-like systems formed by a negative muon and a nucleus. By measuring the 2S-2P energy splitting in muonic hydrogen, muonic deuterium and muonic helium, we have been able to deduce the p, d, [3]He and [4]He nuclear charge radii to an unprecedented accuracy. These radii provide benchmarks for hadron and nuclear theories, lead to precision tests of bound-state QED in regular atoms and to a better determination of the Rydberg constant.

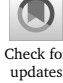
## 21.1 Introduction

Some energy levels of light, hydrogen-like muonic atoms are extremely sensitive to the influence of nuclear properties, such as the nuclear charge and magnetization distributions, and the nuclear polarizability. This makes laser spectroscopy of these states a unique tool for precision determination of these nuclear properties.

Of particular significance is the first excited 2S state in these H-like atoms. First, the 2S state has a large overlap of the muon wave function with the nucleus. Because of the large muon mass, $m_\mu \approx 200\, m_e$, the wave function overlap is about $200^3 \approx$ a few million times larger for muonic atoms, compared to the corresponding electronic atom. This results in a million-fold enhanced shift of the 2S state due to nuclear size effects. Second, in these light muonic atoms, the energy splitting to the neighboring 2P state is only on the order of 1 eV making the Lamb shift(2S-2P energy splitting) accessible to pulsed infrared lasers. And third, the 2S state is metastable.

The various contributions to the Lamb shift $(2S - 2P_{1/2})$ energy differences in $\mu$p, $\mu$d, and $\mu^4$He$^+$ are [1–3]:

$$\Delta E(\mu \text{p}) = \ 206.0336(15) + 0.0332(20) - 5.2275(10) \times r_{\text{p}}^2 \qquad (21.1)$$

$$\Delta E(\mu \text{d}) = \ 228.7767(10) + 1.7449(200) - 6.1103(3) \times r_{\text{d}}^2 \qquad (21.2)$$

$$\Delta E(\mu^4 \text{He}) = 1668.489(14) + 9.201(291) - 106.220(8) \times r_{\alpha}^2 \,, \qquad (21.3)$$

in units of meV when the charge radii $r_X$ are measured in fm, with the $\mu$d equation corrected for nuclear effects calculated only recently [4,5]. Here, the first term is the sum of the "pure" QED effects, the last term is the finite nuclear charge radius effect, and the second term is the remaining nuclear structure effects (elastic and inelastic two- and three-photon exchange, 2PE and 3PE, respectively) [6–12].

## 21.2 The principle of the experiment

The measurement of the 2S-2P transition in these light muonic atoms is based on pulsed laser spectroscopy. Low-energy muons $(\mu^-)$ with a kinetic energy of about 1 keV are stopped in a $(H_2, D_2, He)$ gas target at low pressure (1-2 mbar) and room temperature, forming the corresponding muonic atoms ($\mu$p, $\mu$d, $\mu$He$^+$) in highly excited states with a principal quantum number around $n \approx \sqrt{m_\mu/m_e} \approx 14$. At this low gas pressure, about 99% of the muons then cascade to the 1S ground state within about 100 ns, while the remaining 1% ends up in the metastable 2S state [13, 14]. The 2S state is metastable, because further fast radiative E1 deexcitation is not possible and two-photon deexcitation is slow for these light nuclei. Thus, for low enough gas pressures of $\sim 1$ mbar, only collisional processes with surrounding gas atoms/molecules limit the 2S lifetime to $\tau_{2S} \approx 1\,\mu$s [14, 15]. This lifetime is suitable for pulsed resonant laser excitation to the neighboring 2P state, which quickly de-excites to the 1S ground-state via emission of a Lyman-$\alpha$ X-ray. The detection of this X-ray in time coincidence with the laser light is used to signal a successful laser transition. The resonance is observed by plotting the number of X-rays versus laser frequency.

The experimental setup is based on five main building blocks: a muon beam line delivering negative muons with keV kinetic energy, a detector for these muons based on a set of ultra-thin carbon foils providing a trigger signal for the laser, a laser system capable of delivering high-energy pulses within a short time upon a trigger, a multi-pass optical cavity enhancing the laser fluence at the position of the muonic atoms, and a detection system for the muonic Lyman-$\alpha$ X-rays of a few keV with good energy and time resolutions.

The design of the experiment is dominated by the stochastic arrival time of the muon, the short lifetime of the 2S state, the required very low target gas pressure, and the large laser fluence needed to drive the muonic atom transitions. Muons with energies of few keV stop in a 20 cm long gas target. The low-energy beam line delivers about 500/s detected low-energy muons, each of them triggering the laser system that provides pulses to excite the 2S-2P transition with delay of about 1 $\mu$s.

Due to the 200-times smaller size than regular atoms, muonic atoms have small matrix elements for optical excitation. In conjunction with the short lifetime of the 2S state, the large muon stopping volume (elongated target with size of $7 \times 20 \times 200$ mm$^3$) and the peculiar wavelength of the transition (e.g. 6.0 $\mu$m for $\mu$p), this sets severe requirements for the laser system and the enhancement cavity.

## 21.3 The low-energy beamline

A schematic diagram of the experimental setup is given in Figure 21.1. The low-energy muon beam line was realized at the $\pi$E5 secondary beamline tuned to a momentum of 102 MeV/$c$

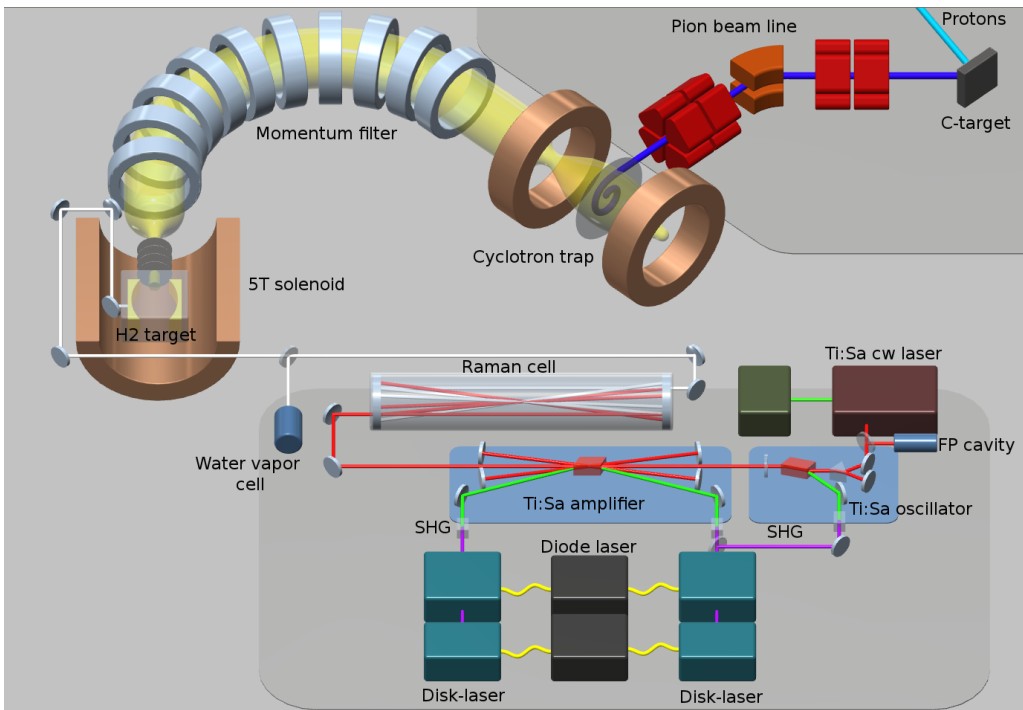

Figure 21.1: Experimental setup used to measure the 2S-2P transitions in $\mu$p.

of the HIPA accelerator at the Paul Scherrer Institute. The negative pions transported by the secondary beam line were injected at a rate of $10^8\,\text{s}^{-1}$ into a cyclotron trap (CT) [16, 17] made of two superconducting 4 T coils. Muons from backwards-decaying pions with energies of a few MeV are confined in the magnetic bottle formed by the two coils. While confined in the trap, the muons slow down by repeatedly passing a 160 nm thick Formvar foil coated with Ni installed in the trap mid-plane. For sufficiently low kinetic energy (around 20 keV), the longitudinal momentum imparted by the –20 kV applied at the foil brings the muon momentum into the loss cone of the trap.

The muons escaping axially from the CT are transported into a region of lower background using a system of 17 coils forming a 0.15 T toroidal magnetic field. This toroidal field also acts as a momentum filter separating the charged particles in the vertical direction according to their momentum. After passing a collimator, which selects muons with the adequate momentum, the muon beam is focused into a 5T solenoid where the gas target is located. The focusing effect caused by the fringe field of the solenoid results in a beam of about 20 mm diameter with kinetic energy of about 20 keV. Before the muons enter the target with a rate of about $500\,\text{s}^{-1}$, and a transverse size of $20 \times 7\text{mm}^2$ (after collimation), they cross several 4 $\mu$g/cm$^2$ carbon foils that are held at high voltage as shown in Figure 21.2. The energy loss occurring in these foils reduces the kinetic energy of the muons to a few keV and frictional cooling [18] reduces their energy spread. The muons crossing the foils also release electrons, which are accelerated by the high voltage applied to the foils, separated from the muon using an $E \times B$-filter and detected in a thin plastic scintillator. This electron signal is used to signal the entering muon providing the trigger for the laser and the DAQ systems.

After crossing the target entrance window of 4 $\mu$g/cm$^2$ thickness, the muons slow down and efficiently (about 80% for 2 mbar pressure) stop in the 20 cm long gas target and form muonic atoms.

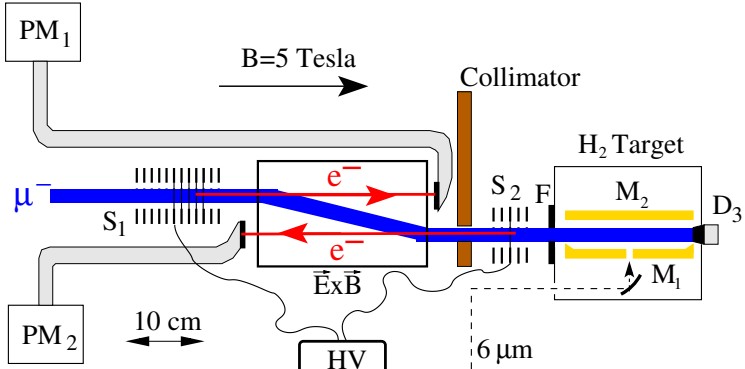

Figure 21.2: Muons are detected by electron emission from two "stacks" of ultra-thin carbon foils before they stop in the gas target. An $\vec{E} \times \vec{B}$ drift region separates the muons from the ejected electrons.

## 21.4 The laser system and the cavity

The laser system for the 2S-2P measurements has to deliver pulses of 0.15 mJ energy tunable from a wavelength of 5.5 to 6.0 $\mu$m for $\mu$p and $\mu$d [19], and of 10 mJ tunable from 800 to 970 nm for $\mu^3$He$^+$ and $\mu^4$He$^+$. Moreover the laser system has to respond to a stochastic trigger and have a short latency time ($\lesssim 1\,\mu s$), i.e., a short delay between trigger and pulse delivery. Each detected muon that enters the target triggers the laser system, which has to provide the pulses before the 2S state has decayed.

To achieve the needed short latency time and large pulse energy, the laser system starts with two thin-disk lasers (TDL) [20] where the energy is continuously stored in the active medium through continuous wave (cw) pumping with commercial diodes of kW optical power at 940 nm. Each TDL consists of a Q-switched oscillator followed by a multi-pass amplifier. To further reduce the delay time, the oscillator operates in pre-seeding mode prior to the trigger, i.e. in cw-mode at low power close to threshold. The laser cavity is closed when triggered, so that a rapid pulse buildup can start from the circulating laser photons. Cavity dumping is used to extract the pulses which are subsequently sent to the multi-pass amplifier.

The frequency-doubled pulses of the TDL are used to pump a Ti:Sapphire oscillator-amplifier system. The Ti:Sapphire (Ti:Sa) oscillator is injection-locked by a single-frequency master cw Ti:Sapphire laser that is tunable in frequency. For $\mu$He, the pulses of the Ti:Sa laser were used directly to drive the 2S-2P transitions, while for the $\mu$p and $\mu$d measurements the Ti:Sa pulses needed to be frequency-shifted to the 6 $\mu$m region using three Stokes shifts in a Raman cell filled with 15 bar of $H_2$ gas.

To enhance the laser fluence at the muonic atom position that are distributed over a volume of about $7 \times 20 \times 200$ mm$^2$, the laser light is coupled into a multipass cavity through a 0.6 mm diameter hole. The multipass cavity consists of two long mirrors as shown in Fig. 21.3. It is capable of illuminating a large volume extended in longitudinal direction from a transverse direction [21]. The cylindrical mirror confines the injected light in the vertical direction, while the other mirror, formed by a flat central substrate with two cylindrical end-pieces, confines the light in horizontal (longitudinal) direction. The injected light confined within these two mirrors reflects many times (from 500 to 1000 depending on the laser wavelength) between the two optical surfaces homogeneously illuminating the muon stop volume and enhancing the laser intensity.

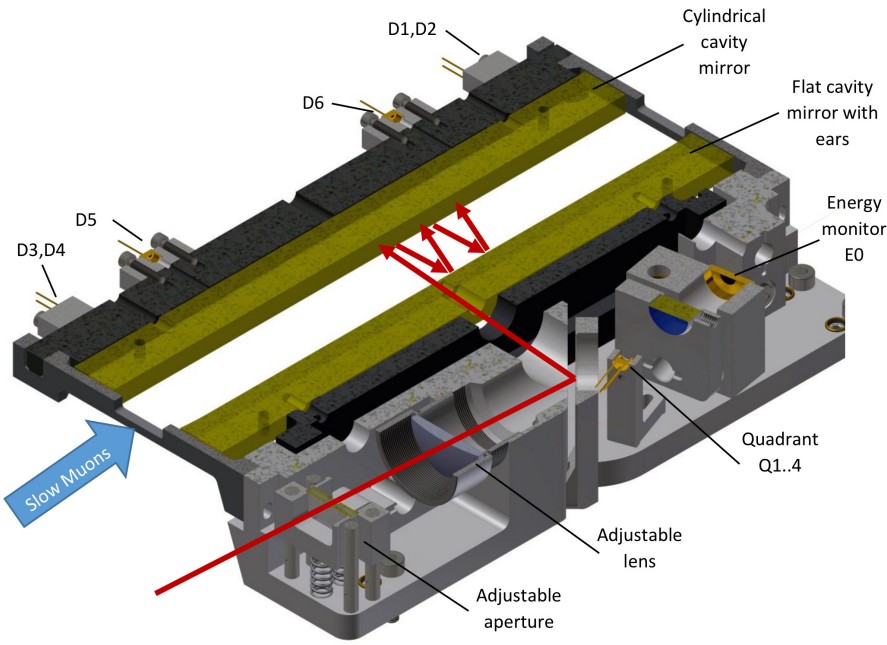

Figure 21.3: The multipass laser cavity used for efficient illumination of the large muon stop volume. The laser beam (red) enters through a hole with a diameter of only 0.6 mm, and bounces between the 2 elongated mirrors to fill the whole cavity volume. One long cylindrical mirror ensures vertical confinement of the light, while the other flat mirror has cylindrical "ears" attached at the ends that result in horizontal confinement [21].

## 21.5 The detectors

The X-ray detection system consists of two linear arrays, each with 10 large area avalanche photodiodes (LAAPDs) of $14 \times 14$ mm$^2$ active area read out with charge sensitive pre-amplifiers. The two detector–pre-amplifier arrays are mounted in the 5T magnetic field above and below the muon stopping volume, resulting in about 25% geometrical acceptance. The energy resolutions at $-30 \pm 0.1°$C are 27% and 16% FWHM for K$_\alpha$ photons at 1.9 keV ($\mu$p) and 8.2 keV ($\mu$He), respectively. The LAAPDs also detect the Michel electrons from muon decays. To improve the electron detection efficiency four plastic scintillators are placed around the target.

The LAAPDs signals were recorded during data taking with waveform digitizers, allowing to reject pile-up events, to disentangle events where the X-ray is followed by the electron from muon decay, and to reject noisy events. Waveform analysis could distinguish between X-rays and electrons from muon decay [22], and improved the energy and time resolutions.

## 21.6 Measurements and results

In total, ten transition frequencies in $\mu$p [23,24], $\mu$d [25], $\mu^4$He [26] and $\mu^3$He were measured (manuscript on $\mu^3$He is in preparation). A low background rate of 1 event/h was observed in all these measurements as due to the use of a continuous muon beam. With only a single muon at a time in the apparatus, the data analysis rejected events with multiple signals. The single-muon event analysis also allowed the detection of the muon-decay electron following a Lyman-$\alpha$ X-ray resulting in a strong suppression of background events. The detection of this decay-electron and related background suppression favors cw over pulsed muon beams. However, this comes at a price: the laser has to cope with large repetition rates, with a stochastic

trigger and has to have a small latency time between muon trigger and pulse delivery. The development of the adequate laser technologies was one of the main challenges of these experiments.

As a result of the successful background suppression, signal to background ratios (at resonance) of about 5 have been obtained. Signal rates of 6 events/h were observed on resonance, so that the measurement of each transition required about one week of data taking. The centroid positions were deduced for the measured resonances with accuracies between $\Gamma/10$ and $\Gamma/20$, where $\Gamma$ is the FWHM linewidth of the resonances ($\Gamma \approx 20$ GHz for $\mu$p, $\Gamma \approx 320$ GHz for $\mu\text{He}^+$). The 'pure''(free from hyperfine splitting effects) Lamb shifts [23–26], obtained from several measurements, are:

$$\Delta E(\mu\text{p}) = 202.3706\,(19)_{\text{stat}}\,(12)_{\text{syst}}\ \text{meV} = 202.3706\,(23)_{\text{total}}\ \text{meV} \tag{21.4}$$

$$\Delta E(\mu\text{d}) = 202.8785\,(31)_{\text{stat}}\,(14)_{\text{syst}}\ \text{meV} = 202.8785\,(34)_{\text{total}}\ \text{meV} \tag{21.5}$$

$$\Delta E(\mu^4\text{He}^+) = 1378.521\,(46)_{\text{stat}}\,(12)_{\text{syst}}\ \text{meV} = 1378.521\,(48)_{\text{total}}\ \text{meV}\,. \tag{21.6}$$

The experimental accuracies are all limited by statistical uncertainties. The experiment has small sensitivity to typical atomic physics systematic errors, such as Doppler, Stark and even the Zeeman shifts in the 5T field, and laser frequency calibration.

By comparing these measurements to the corresponding theoretical predictions (21.1)–(21.3), we obtain the following nuclear charge radii

$$r_{\text{p}} = 0.84087\,(26)_{\text{exp}}\,(29)_{\text{theo}}\ \text{fm} \tag{21.7}$$

$$r_{\text{d}} = 2.12718\,(13)_{\text{exp}}\,(89)_{\text{theo}}\ \text{fm} \tag{21.8}$$

$$r_{\alpha} = 1.67824\,(13)_{\text{exp}}\,(82)_{\text{theo}}\ \text{fm}\,. \tag{21.9}$$

With the exception of $\mu$p, where the theoretical and experimental uncertainties are similar, the theoretical uncertainty of the calculated nuclear 2PE and 3PE contributions presently limit the extraction of the nuclear charge radii from these measurements.

## 21.7  Impact

The proton radius extracted from $\mu$p [23, 24] is an order of magnitude more precise than previous determinations. There is a large, unexpected discrepancy with the values from both electron scattering [38] and H spectroscopy: this is the "proton radius puzzle" [39, 40]. This has triggered various theoretical efforts including refinement of bound-state QED calculations for the atomic energy levels [41–46], refinement of techniques to extract the proton charge radius from scattering data [27, 47–53], investigations on the proton structure [8–12], investigation of beyond standard model physics [54–57], and refinements of laser spectroscopy systematic effects such as quantum interference [58, 59]. These investigations have considerably advanced our understanding but have been unable to explain the observed discrepancy. At the same time various experimental activities were initiated ranging from spectroscopy of hydrogen atoms, hydrogen molecules, electron and muon scattering, laser spectroscopy of Muonium and Rydberg atoms. Recently, several of these experimental efforts produced new results: all of them but one in excellent agreement with the proton radius value as extracted from muonic hydrogen and in some tension with previous hydrogen and electron-scattering results [29–33].

By assuming the correctness of the proton radius as extracted from muonic hydrogen, the Rydberg constant $R_\infty$ has to be revised. Using the precise value of the proton radius from muonic hydrogen its relative uncertainty is decreased to $8 \times 10^{-13}$, which is the most precise value for a fundamental constant.

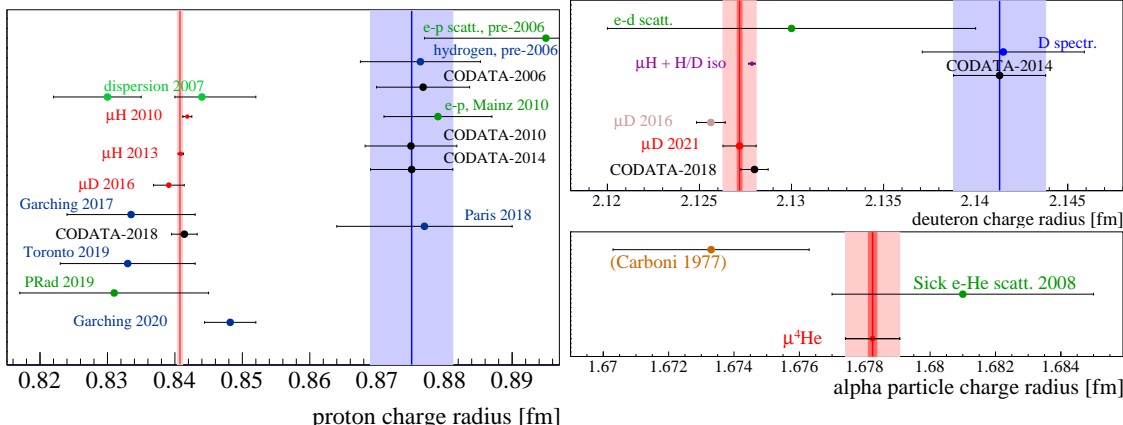

Figure 21.4: The charge radii from muonic atoms and other methods. For the radii from muonic atoms we separate the experimental uncertainties (dark red bands) from the theory uncertainties arising mainly from the 2PE contribution (lighter red band). For the proton (left), historical values and the 2010 Mainz A1 result [27] agree on a value around 0.88 fm, except for dispersion fits [28]. Muonic hydrogen [23, 24] and muonic deuterium [25] require a smaller radius around 0.84 fm. Whereas a new result from hydrogen 1S-3S (Paris 2018 [29]) seems to favor the larger radius, more recent measurements from hydrogen spectroscopy H(2S-4P) (Garching 2017 [30]), H(2S-2P) (Toronto 2019 [31]), and H(1S-3S) (Garching 2020 [32]) as well as a low-$Q^2$ e-p scattering experiment by the PRad Collaboration [33] favor the smaller radius. CODATA has now accepted the smaller radius.

For the deuteron (right top), older laser spectroscopy in atomic D favor the larger radius around 2.14 fm, but the smaller *proton* radius from muonic hydrogen, together with the isotope shift of the 1S-2S transition in regular H and D from Garching [34] yield a smaller radius of 2.12 fm. The value from muonic deuterium [25] has recently been brought into agreement with the latter more precise value by improved nuclear theory [4,5,35]. Elastic electron-deuteron scattering [36] cannot resolve the difference.

For the alpha particle, no value from regular atoms exists. Elastic e-He scattering [37] is five times less accurate than the muonic value. The historical $\mu$He value from Carboni is wrong.

The $r_\alpha$ value extracted from $\mu^4$He$^+$ [26] is in excellent agreement with the world average value from elastic electron scattering [37] but almost 5 times more precise. Hence it serves as a benchmark for few-nucleon theories [6, 60], for lattice QCD calculations and for elastic electron-He scattering. It serves also as an anchor point for isotopic shift measurements opening the way to improved values of the $^3$He, $^6$He and $^8$He nuclei, and can be used to test higher-order bound-state QED contributions to an unprecedented sensitivity when combined with measurements in regular He$^+$ and He atoms.

## 21.8 Outlook

As a next step, the CREMA collaboration is addressing the hyperfine splitting of the ground state in muonic hydrogen. The goal is to measure this transition with 1-2 ppm precision from which the 2PE contribution can be obtained with $10^{-4}$ relative accuracy. The extracted 2PE contribution can be then compared to predictions from chiral perturbation theory (chPT) or from data-driven (proton structure functions and form factors) dispersion relations [11,61,62].

In this experimental effort, an improvement in laser technology is underway. The improved technology will also open the way for an improved measurement of the 2S-2P transitions: a factor of 5 improvement seems to be possible for all four muonic atoms.

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
