# Peer review of "Laser spectroscopy of light muonic atoms and the nuclear charge radii"

_SciPost Physics Proceedings, doi:SciPost Phys. Proc. 5, 021 (2021)_

## Round 1 · Referee Report · Adrian Signer (Referee 1) · 2021-6-22

Report
We (the editors Cy Hoffman, Klaus Kirch, Adrian Signer) had the
opportunity to review an earlier draft of the article and were in
communication with the authors before the submission. All our comments
and suggestions have been taken into account. Hence, we think the
paper can now be published in the current form.
opportunity to review an earlier draft of the article and were in
communication with the authors before the submission. All our comments
and suggestions have been taken into account. Hence, we think the
paper can now be published in the current form.

---

## Round 1 · Referee Report · Anonymous (Referee 2) · 2021-7-13

Report
Laser spectroscopy of light muonic atoms, starting with the pristine muonic hydrogen system and expanding to muonic deuterium and Helium ions has been a scientific highlight of the PSI research program. It allows for precision measurements which are very difficult to match by any other method, as the tightly bound muon probes the electromagnetic structure of the nucleon and light nuclei with a few million times larger overlap than the electrons of ordinary atoms. The surprising disagreement with previous measurements on the proton charge radius, dubbed the proton radius puzzle, caught the interest of the physics community at large, including atomic physics as well as nuclear and particle physics. It stimulated a rich research program both in theoretical developments and a new generation of precision measurements and changed the previously accepted value of the Rydberg Constant. After years of debate a largely consistent picture is emerging with the muonic atom data not only holding up, but defining a new precision standard in this field.
This paper presents a comprehensive overview of the scientific questions, the experimental method, which is based on several remarkable technical achievements, and the impact of the results on various fields of physics. It is formulated well and includes extensive references for more detailed questions. The paper can be published as it, as it clearly exceeds all the required criteria for publication.
Below I add some minor remarks for the author's consideration.
Fig 21.3: In the way the CAD of the cavity is cut, the two sides look the same. So it is hard to comprehend the difference between the cylindrical mirror side and the flat mirror with the ears side. Perhaps this figure could be improved.
Line 147. It would be helpful to state in the preceding paragraph that the Michel electrons are detected by the LAAPDs.
Fig. 21.4: Define the lighter/darker bands and colors early on in the figure caption. Are they consistently used? Carboni is colored in blue, though this was a muonic experiment, Sick in magenta, though probably this was an e-scattering evaluation.
3He measurements are mentioned, but no results presented. Maybe comment on the status of this analysis.
This paper presents a comprehensive overview of the scientific questions, the experimental method, which is based on several remarkable technical achievements, and the impact of the results on various fields of physics. It is formulated well and includes extensive references for more detailed questions. The paper can be published as it, as it clearly exceeds all the required criteria for publication.
Below I add some minor remarks for the author's consideration.
Fig 21.3: In the way the CAD of the cavity is cut, the two sides look the same. So it is hard to comprehend the difference between the cylindrical mirror side and the flat mirror with the ears side. Perhaps this figure could be improved.
Line 147. It would be helpful to state in the preceding paragraph that the Michel electrons are detected by the LAAPDs.
Fig. 21.4: Define the lighter/darker bands and colors early on in the figure caption. Are they consistently used? Carboni is colored in blue, though this was a muonic experiment, Sick in magenta, though probably this was an e-scattering evaluation.
3He measurements are mentioned, but no results presented. Maybe comment on the status of this analysis.

---

## Round 2 · Author Response

Dear Editor

we agree with the very appropriate comments of the referee and we have modified the text accordingly.

Many thanks
and best regards
Aldo Antognini

---

## Round 2 · List of Changes

1) Referee
Below I add some minor remarks for the author's consideration.
Fig 21.3: In the way the CAD of the cavity is cut, the two sides look the same. So it is hard to comprehend the difference between the cylindrical mirror side and the flat mirror with the ears side. Perhaps this figure could be improved.

Our reply:
The CAD drawings account for the correct geometry of the mirrors. The curvatures are relatively small so not really visible in the figure. Yet, the details about the cavity can be found in Ref. 21 which is mentioned in the caption. For this reason we prefer not to modify the figure.

2) Referee:
Line 147. It would be helpful to state in the preceding paragraph that the Michel electrons are detected by the LAAPDs.

Our reply:
We agree with the referee. Hence we added following sentences at lines 137-138:
"The LAAPDs also detect the Michel electrons from muon decays. To improve the electron detection efficiency four plastic scintillators are placed around the target."

3) Referee
Fig. 21.4: Define the lighter/darker bands and colors early on in the figure caption. Are they consistently used? Carboni is colored in blue, though this was a muonic experiment, Sick in magenta, though probably this was an e-scattering evaluation.

Our reply:
We agree with the referee. The definition of the red shaded bands has been shifted at the beginning of the caption.
We have also modified the color with the He charge radii as correctly pointed out.

4) Referee:
3He measurements are mentioned, but no results presented. Maybe comment on the status of this analysis.

We agree with the referee. For this reasons at line 144-145 we added:
"In total, ten transition frequencies in μp [23,24], μd [25], μ4He [26] and μ3He were measured (manuscript on μ3He is in preparation)."

---

## Editorial Decision

published